# Study protocol for a randomised, controlled platform trial estimating the effect of autobiographical Memory Flexibility training (MemFlex) on relapse of recurrent major depressive disorder

Caitlin Hitchcock,[1] Siobhan Gormley,[1] Cliodhna O'Leary,[1] Evangeline Rodrigues,[1] Isobel Wright,[1] Kirsty Griffiths,[1] Julia Gillard,[1] Peter Watson,[1] Emily Hammond,[1,2] Aliza Werner-Seidler,[3] Tim Dalgleish[1,4]

[1]Medical Research Council Cognition and Brain Sciences Unit, University of Cambridge, Cambridge, UK
[2]Psychology, University of Exeter, Exeter, UK
[3]The Black Dog Institute, Sydney, Australia
[4]Cambridgeshire and Peterborough NHS Foundation Trust, Cambridge, UK

**Correspondence to**
Dr Caitlin Hitchcock;
Caitlin.hitchcock@mrc-cbu.cam.ac.uk

## ABSTRACT

**Introduction** Major depressive disorder (MDD) is a chronic condition. Although current treatment approaches are effective in reducing acute depressive symptoms, rates of relapse are high. Chronic and inflexible retrieval of autobiographical memories, and in particular a bias towards negative and overgeneral memories, is a reliable predictor of relapse. This randomised controlled single-blind trial will determine whether a therapist-guided self-help intervention to ameliorate autobiographical memory biases using Memory Flexibility training (MemFlex) will increase the experience of depression-free days, relative to a psychoeducation control condition, in the 12 months following intervention.

**Methods and analysis** Individuals (aged 18 and above) with a diagnosis of recurrent MDD will be recruited when remitted from a major depressive episode. Participants will be randomly allocated to complete 4 weeks of a workbook providing either MemFlex training, or psychoeducation on factors that increase risk of relapse. Assessment of diagnostic status, self-report depressive symptoms, depression-free days and cognitive risk factors for depression will be completed post-intervention, and at 6 and 12 months follow-up. The cognitive target of MemFlex will be change in memory flexibility on the Autobiographical Memory Test- Alternating Instructions. The primary clinical endpoints will be the number of depression-free days in the 12 months following workbook completion, and time to depressive relapse.

**Ethics and dissemination** Ethics approval has been granted by the NHS National Research Ethics Committee (East of England, 11/H0305/1). Results from this study will provide a point-estimate of the effect of MemFlex on depressive relapse, which will be used to inform a fully powered trial evaluating the potential of MemFlex as an effective, low-cost and low-intensity option for reducing relapse of MDD.

**Trial registration number** NCT02614326.

## Strengths and limitations of this study

► This early-stage platform trial seeks to establish an estimate of effect size to power a later-stage trial.
► The intervention condition will be compared with an active control condition.
► The hard-copy workbook format limits the ability to track completion of the intervention sessions in real time.
► The absence of booster sessions following workbook completion may limit the long-term efficacy of the intervention.

## INTRODUCTION

Depression is a recurrent and pervasive condition which dramatically reduces quality of life for the individual and costs the health system in the UK an estimated £7.5 billion per year.[1] Although psychological approaches to treating depression are effective in reducing current symptoms,[2] relapse rates are high. On average, those with major depressive disorder (MDD) will experience 10 separate depressive episodes across their lifespan,[3] and symptom severity increases with each experienced episode.[4] As such, reducing depressive relapse is a key challenge for health systems. Broadly, depressive risk is increased by the chronic use of maladaptive cognitive styles which promote depressogenic thoughts, attributions and beliefs,[5] thus, intervention to reduce the chronicity of these cognitive styles may thereby break the vicious cycle of depression. Importantly, as cognitive risk factors for depression remain evident when depressive symptoms remit, it may be possible to complete interventions to shift these factors when the individual is psychologically well, and more able to engage in therapy. Further exploration of low-cost and low-intensity options for reducing these cognitive risk factors may have important implications for

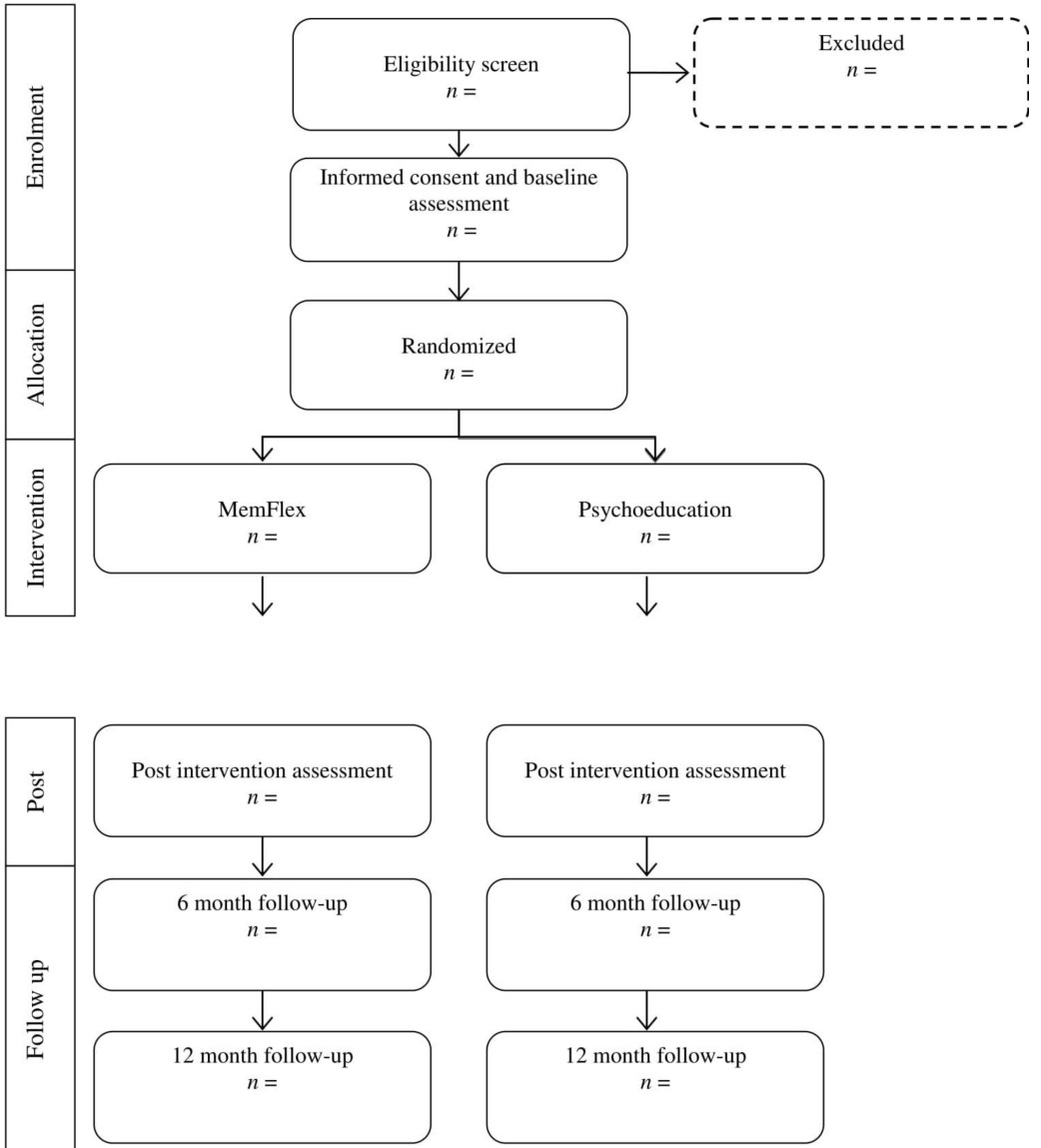

**Figure 1** Flow chart of study design (Consolidated Standards of Reporting Trials diagram). All outcome and process measures will be completed at all time points. MemFlex, Memory Flexibility training.

limiting depressive relapse and subsequent costs for the individual and healthcare system. This trial will investigate the efficacy of a low-intensity programme, Memory Flexibility training (MemFlex), which aims to reduce one such risk factor—inflexible retrieval of autobiographical memories.

Autobiographical memory, that is, memory of personal life experiences, is integral in forming one's sense of self,[6] and disruptions in autobiographical memory increase risk of depression (for review see ref[7]). Autobiographical information is stored in a fluid manner, and can therefore be retrieved in a variety of formats, including as specific, single incident events that are rich in detail (eg, when I forgot my purse and couldn't buy my train ticket), or as generalisations that summarise consistency across a category of events (eg, catching the train to work every day). Memory flexibility refers to the ability to successfully navigate the autobiographical information database to deliberately retrieve memories varying in detail and emotional valence. In depression, flexible movement within the autobiographical information database is compromised,[8 9] resulting in relative difficulties in retrieving both general and specific memories.[8 10 11] This can create functional impairment as both types of memory are necessary in daily cognitive skills; for example, specific memories can help to guide problem solving[12] while general memories encode regularities of experiences that can be used to inform judgements of the self, world and others.[13 14] Impaired recall of specific memories is a key predictor of the first onset of depression[15] and also of future depressive episodes in those with a history of recurrent depression (for meta-analysis see ref [16]), although comparable studies examining the predictive effects of impaired generality or flexibility have yet to be carried out.

A core feature of reduced flexibility is the chronic over-retrieval of autobiographical memories that are: (1) negative in emotional valence and (2) overgeneral in nature. Regarding valence, depressive risk is increased by a habitual and inflexible inclination towards negative memories, and subsequently, a neglect for positively valenced and neutrally valenced memories.[17] However, it is not simply that negative information is retrieved more readily, but that when positive information is recalled, it is less vivid[18] and therefore has an attenuated impact on affect, reducing the emotional benefits that are gained from recalling positive memories.[19] In this way, biased memory retrieval can impair the ability to repair transient low mood.[20] In addition, the absence of vivid, positive memories of oneself and a continual emphasis on negative autobiographical information reinforces the negative self-schemata that maintain and promote depression.[21]

Difficulties in intentional autobiographical retrieval also increases risk for depression. A tendency towards general memory recollection and reduced retrieval of specific memories is evident in those with current depression, however, the overgeneral retrieval style does not remit with depressive symptoms,[11] suggesting that reduced specific recall is a durable cognitive marker for depression[16] that may not be shifted by current psychological treatments,[22] though this has not yet been well explored. Repeated recall of overgeneral summaries may promote depression by reinforcing overly general (negative) attributions regarding the self (eg, I always fail) and the world (eg, the world is an unfair place). Previous research has also demonstrated that impaired retrieval of specific memories impacts a number of intermediate factors that promote depression. Abstract and overgeneral memory recall intensifies rumination,[23] which in turn creates negative affect, impairs mood regulation, and is itself a risk factor for recurrent depression.[24] Impaired retrieval of specific memories also reduces the ability to establish intimacy in social communication,[25] impairs problem solving[26] and limits the ability to appropriately restrain the overly general, negative self-judgements that are characteristic of depression.[27]

As reduced flexibility in the use of negative and overgeneral memory retrieval styles both independently promote depression symptoms[8 16] and may also interact with other risk factors (eg, stress[28]); to trigger a depressive episode, intervening to reduce bias and improve memory flexibility may thereby reduce depressive risk. There is some evidence to suggest a positive impact of targeting these factors individually, such as by reducing overgeneral memory through memory specificity training.[29–31] However, simultaneously targeting these factors may reduce compounded effects, and promote flexibility in the use of other depressogenic cognitive styles.[32] In addition, explicitly training improved recall of positive, general memories (eg, of kindness, determination, caring relationships) may help to support reinitiation of the overly general, positive self-judgements that are associated with mental health (for review, see ref [33]).

To this end, we have previously piloted a novel intervention which aims to improve cognitive flexibility in those experiencing recurrent depression.[34] MemFlex uses autobiographical memory as a modality to train negative and overgeneral processing styles. The workbook-based programme is presented in eight self-guided sessions completed over a 1-month period, and involves completion of memory retrieval exercises that aim to improve concrete, specific processing, thereby reducing overgeneral, abstract processing and improve access to neutral and positive emotional material, thereby reducing negative bias. In an initial uncontrolled trial with those in remission from MDD, MemFlex was effective in reducing overgeneral memory (d=0.48), rumination (d=0.29), and cognitive avoidance (d=0.18), and also improved social problem solving (d=0.55) from preintervention to postintervention.[34] An ongoing randomised controlled trial is comparing the effect of MemFlex to an active psychological intervention for those experiencing a current major depressive episode (see ref[35] for trial protocol). However, results from the uncontrolled trial suggest that MemFlex may not only prove beneficial for current symptoms of depression, but also for reducing cognitive risk factors for depressive relapse.

Preliminary evidence of an effect on multiple depressive risk factors provides a solid foundation for a larger trial of MemFlex determining the efficacy of the programme in relapse prevention, against an active control group.[36] In line with recommendations for the phase-based development and evaluation of novel interventions, we will complete an early-stage, randomised controlled platform trial with the primary aim of estimating the likely effect size of MemFlex on measures of depressive relapse, which can then be used to inform a later-stage, fully powered trial. This study will therefore estimate the effect of MemFlex on the recurrence of depression over a 12-month period, relative to an active control condition—psychoeducation. We hypothesise effect sizes in favour of MemFlex, such that individuals completing MemFlex when they are currently remitted from depression will experience a greater number of depression-free days in the 12 months following workbook completion and demonstrate a longer period of time until depressive relapse, compared with those who complete psychoeducation.

## METHODS AND ANALYSIS
### Study design
The design is a single-blind, participant-level randomised controlled trial comparing MemFlex to psychoeducation. All participants will provide informed consent to complete a 4-week programme, consisting of an initial face-to-face session, followed by eight self-guided workbook sessions presenting either the psychoeducation or MemFlex materials. Participants will be assessed at

baseline, postintervention, at 6 months follow-up and at 12 months follow-up.

## Participants and recruitment

Inclusion criteria are a minimum age of 18 years and a primary diagnosis of MDD, as assessed using the Structured Clinical Interview for DSM Disorders (SCID[37]). On study entry, participants must not be currently experiencing a major depressive episode. That is, participants must be in partial or full remission from MDD, as indexed by the Longitudinal Interval Follow-up Evaluation (LIFE[38]) for the SCID. Additional exclusion criteria are prior completion of any autobiographical memory-based training programme, experience of another current mood disorder, psychosis, personality disorder or current drug, or alcohol dependence. Those experiencing significant cognitive impairment, as assessed through self-report (via an open-ended question asking whether the individual has ever experienced an intellectual disability or traumatic brain injury), will also be excluded. Participants may continue with any current recovery-maintenance medication or psychological interventions (eg, booster sessions, mindfulness practice) while participating in the study.

We will recruit through three avenues in Cambridge, UK. First, participants will be invited to contact us through advertisements in the general community (eg, posters in general practitioner surgeries) and the local newspaper. All those who contact us will be sent further information about the study, and invited to complete an initial telephone screening. We will also recruit from our database of participants with depression who are interested in taking part in research. All participants on this database have previously completed the SCID and meet criteria for MDD. These participants will be telephoned or emailed to make them aware of the study, and all those who express an interest in participating will complete the LIFE to determine current depression and wider diagnostic status. Finally, we will recruit patients who have expressed an interest in participating in research from local mental health services. Clinical care teams will be made aware of the study, and will be encouraged to ask their patients if they would like to take part in research. If the patient expresses an interest in research, their clinical care team will pass on our information sheet.

## Participant allocation

Random allocation to condition using computer-generated, quasi-random numbers will completed by the trial statistician (PW), who is blind to study objectives. Once a participant has provided informed consent and completed the preintervention assessment, the researcher completing the initial session will open a sealed, opaque envelope to reveal condition allocation. This process is demonstrated in the Consolidated Standards of Reporting Trials (CONSORT) diagram (see figure 1).

## Interventions

### MemFlex

We have previously piloted the MemFlex programme with individuals in remission from depression.[34] The intervention will be delivered as per the pilot trial and the trial evaluating the intervention in those with current depression.[35] The programme aims to improve the flexibility of autobiographical retrieval processes, to improve the ease with which individuals can access and move between specific and general personal memories, and to train away a bias towards negative memories. This is achieved through repeated completion of cued recall tasks which use positive and neutral cues. The training exercises are introduced in an initial face-to-face session with a researcher, and the participant is then required to complete two times weekly sessions of training over a 4-week period. During the initial session, the researcher provides information about the role of autobiographical memory in depression, including the different types of autobiographical memories (eg, specific event memories, and the generalised themes that link together a series of events) and their presumed role and importance in daily life. The rationale for MemFlex is then introduced. The researcher then assists the individual to complete practice exercises, and provides feedback and encouragement to ensure that the individual understands the tasks. The session lasts between 45 and 60 min. The participant will also receive a telephone call from the researcher halfway through the time set to complete the workbook to answer any questions, and address any difficulties the individual may be having with the workbook. The participant is also encouraged to contact the researcher with any questions about the workbook content.

The MemFlex workbook is separated into eight sessions. The early sessions of the workbook focus on reducing the bias towards negative and general memories experienced in depression by promoting the retrieval of both specific and general memories of neutral and positive valence. This involves repeated cued-recall exercises using neutral and positive cues. Next, the workbook encourages the individual to elaborate the specific emotional details attached to positive memories, again using repeated retrieval tasks. This seeks to improve the reduced vividness of positive memories experienced in depression.[18] The final workbook sessions involve completion of exercises that require the participant to explicitly link specific event memories with a general theme, and in reverse, to identify individual event memories that form a general theme. To do this, individuals must explicitly switch between retrieving specific and general memories to the same cue, thereby promoting flexibility in movement between the different memory types. When completed with positive cues, this type of exercise also aims to promote positive generalisations about the self. Exercises are repeated throughout the workbook to

give participants practice in using the different skills. There are additional exercises at the end of the workbook that the participant can choose to do after the study has ended. While participants are encouraged to continue to apply their new skills in their everyday life, completion of the additional workbook exercises is optional.

## Psychoeducation

Participants in the psychoeducation condition will also attend a 45–60 min face-to-face session, followed by the completion of a workbook over a 4-week period. The initial session will provide information on the causes of depressive relapse, and the individual's experience of relapse will be discussed between the researcher and participant. The eight-session workbook will then be introduced.

The psychoeducation workbook is a relapse-focused adaptation of that used in a previous trial for those with MDD.[35] The workbook presents theory on different cognitive and lifestyle factors that are associated with depression which may persist between depressive episodes, and promote the experience of low mood in the future. These include poor sleeping patterns, worry and perfectionism. After the relevant theory is explained, each component of the workbook will offer tips for improving each factor, and an exercise which encourages the participant to reflect on whether these factors occur in their life. Four multiple-choice questions are also asked at the end of each session to ensure participant engagement, and to match the completion of workbook exercises against the MemFlex condition. Interventions are matched for time burden.

## Treatment integrity

Both workbooks are available from the corresponding authors on request, and will be made available online if their efficacy is demonstrated. Researchers will be trained and supervised by a clinical psychologist (CH) to administer the manualised initial sessions. Sessions will be audio-recorded, and 15% of these will rated for adherence to the manual by a researcher independent from the project. At the end of the initial session, participants will complete the Credibility Expectancy Questionnaire—Patient Version[39] to index any difference in treatment expectations between the two intervention conditions. Progress on the workbook will be checked halfway through the 4 weeks provided for workbook completion. If the individual is behind on workbook completion, any difficulties in completing the workbook will be briefly addressed, and trial management records will be updated with an anticipated finish date. A member of the research team will call the individual the week before the anticipated finish date to set an appointment time for the postintervention assessment. To ensure adherence, the workbook will also be viewed at postintervention to ensure all materials have been attempted. The number of completed sessions will be recorded.

## Measures
### Cognitive target

We anticipate that the MemFlex programme will improve, and maintain improvement in, memory flexibility relative to psychoeducation. We will therefore administer the Autobiographical Memory Test with Alternating Instructions (AMT-AI[8]) at all assessments. The AMT-AI requires individuals to recall either specific or general memories in response to neutral, positive and negative cue words. Participants are asked to retrieve specific memories to a block of six cue words, general memories for a block of six cues and finally a block of 12 cues in which the individual must alternate between retrieval of specific and general memories. The cues are randomised to blocks, and the order in which the blocks are presented is also randomised. The main outcomes are the number and proportion of correct responses for each block (ie, specific, general, alternating).

### Primary clinical outcomes

The primary clinical outcomes are number of depression-free days from postintervention to 12-month follow-up, and number of days until depressive relapse, both according to the LIFE.

### Secondary clinical outcomes

The secondary clinical outcomes are depressive status at 12-month follow-up, again measured using the LIFE and symptoms of depression at 12-month follow-up as indexed on the Beck Depression Inventory II (BDI-II[40]).

### Additional process measures

We have previous evidence to suggest that MemFlex will impact rumination, social problem solving and cognitive avoidance in samples remitted from depression.[34] We will therefore assess whether these factors may mediate the impact of MemFlex on our primary and secondary outcomes. Rumination will be measured using the Rumination Response Scale,[41] cognitive avoidance will be measured using the Cognitive Avoidance Questionnaire[42] and social problem solving will be indexed on the short version of the Means-Ends Problem-Solving (MEPS) task.[43] We will also assess verbal fluency using the Verbal Fluency Task[44] to determine whether the programme impacts the fluent retrieval of verbal information. Each of these measures possesses adequate psychometric properties and has been previously used in evaluation of autobiographical memory interventions.[7 34 35] Parallel forms of these measures will be counterbalanced and administered at all time points.

## Methodological aspects
### Power analysis

As this is an exploratory, early-stage randomised controlled platform trial (in accordance with recommendations for the evaluation of complex interventions; Medical Research Council (MRC), 2000), we aim to generate estimates of the range of likely effect sizes of MemFlex (relative to psychoeducation) on our primary

clinical outcomes. This can then be used to inform a power calculation for a fully powered later-phase trial. Our prior experience with autobiographical memory-based interventions for depression (eg, refs [29 34 35 45]) suggests that 25 participants per condition will provide a plausible range of point estimates. Given the long-term follow-up we implement, we aim to recruit 35 participants in each condition (70 participants in total) to allow for up to 25% attrition.

### Data collection and blinding
Participants will be assessed at baseline, and then again at postintervention, and at 6-month and 12-month follow-ups after the intervention has been completed. All sessions will be completed at the MRC Cognition and Brain Sciences Unit in Cambridge, UK, except in the case that a participant moves away from Cambridge before the completion of all assessments. In these cases, 6-month and 12-month follow-up assessments may be completed over the telephone. As the AMT-AI is unable to be completed over the telephone, it will not be administered at follow-up for these participants. Research staff completing the assessments will be supervised by a clinical psychologist (CH) and inter-rater reliability will be completed for the AMT-AI and MEPS to ensure data quality and protection from bias. Bias will also be protected against at all assessments through use of research staff whom are blind to experimental condition.

### Statistical analysis plan
The trial statistician is blind to study objectives, and will complete all analyses in accordance with CONSORT standards. No interim analyses are planned during data collection. Between-group difference on our primary clinical outcome of number of depression-free days will be analysed using an independent samples t-test, or analysis of covariance if necessary to account for any baseline differences. Time to depressive relapse will be compared between groups using Cox regression survival analysis. Analysis of our cognitive target variable, secondary clinical outcomes and additional process measures will be analysed using analysis of variance (ANOVA) with time (pre, post, 6 and 12 follow-up) as a within-subjects factor, and condition (MemFlex, psychoeducation) as a between subjects factor. Any baseline differences in the cognitive target, symptom severity on the BDI-II, concurrent treatment or demographics will be covaried in analyses. Diagnostic status at 12-month follow-up will analysed using logistic regression. Residual depression symptoms and number of previous episodes (as an index of depression severity), and score on the AMT-AI (as an index of bias severity) at preintervention will be explored as moderators of treatment efficacy by computing product variables that will be analysed using linear regression. Finally, we will explore mechanisms of change by determining whether intervention effects are mediated by change on the AMT-AI and the other process measures.

Indirect effects will be analysed using bootstrapped samples in the Process statistical package. Intent-to-treat analysis will be used for those lost to attrition by using multiple imputation to account for missing data. The number of imputed datasets will approximate the percentage of missing data.[46] Any exploratory, per-protocol analyses will use data from participants who have completed six out of the eight workbook sessions. Exploratory ANOVAs will also be completed to explore between-group difference in pre-to-post intervention change on the separate AMT-AI blocks, to determine whether MemFlex may have a larger impact on specific, categoric or alternating recall.

### Trial management
The trial management committee will be formed by TD, CH and trial statistician (PW). The trial coordinator (CH) will complete data checks to ensure data quality.

### Ethics and dissemination
All individuals will provide informed consent prior to participation, and Medical Research Council, National Health Service and professional ethical guidelines will be adhered to throughout the study. Any necessary amendments to this protocol will be published in the trial manuscript. No restrictions have been placed on the publication of results. In addition to publication of the trial paper, results will be shared in written form to trial participants, and through seminars to clinicians and the general public.

### Safety aspects
In the unlikely occurrence of adverse events, the Medical Research Council's structured risk assessment protocols will be used to manage risk. All risk issues will be managed by the trial coordinator (CH), in consultation with TD, both of whom are clinical psychologists experienced in treatment of depression. Any adverse events will be reported to the research ethics committee, and will be reported in the trial manuscript.

### Confidentiality and access to data
Only the trial coordinator and trial statistician will have access to the final dataset. Data will be securely stored in de-identified form on-site, and in encrypted electronic format. Personally identifiable information will be stored separately to the trial data, and will only be accessible via a dedicated access server. Once the data have been analysed and the trial is published, the final dataset will be deposited in de-identified form into the research unit's online data repository, along with the used statistical code.

### Trial registration
The trial protocol was registered online on 25 November 2015 at ClinicalTrials.gov (identifier NCT02614326), with recruitment planned to open in January 2016. The first participant was recruited on 18 January 2016.

## DISCUSSION

Major depression is predicted to be the largest global burden of disease by 2030.[47] Reducing the recurrence of depression will therefore be a key issue for the coming years. This study will estimate the effect of a novel intervention, MemFlex, which aims to reduce negative and overgeneral memory biases, and thereby risk of depressive relapse. In simultaneously targeting these factors, the programme may reduce both direct and compounded effects on depressive relapse. This trial aims to estimate the range of likely effects of MemFlex, to provide a platform for a later-stage, fully powered trial, and to begin to explore mechanisms of change. If effective, MemFlex may hold promise as a low-cost and low-intensity option for reducing depressive relapse as the self-guided, workbook format of the programme would be simple and cost-effective to provide to individuals on discharge from a treatment service, or potentially as an adjunct to complex interventions such as cognitive-behavioural therapy. As the pressure that depression places on healthcare systems is expected to increase,[48] further development of low-cost relapse prevention programmes will be vital in improving access to psychological services and quality of life for those experiencing chronic and recurrent depression.

**Contributors** CH codeveloped the study design, cowrote the intervention materials and cowrote the manuscript. SG, CO, ER, IW, KG and JG are contributing to data collection for the trial. PW is the trial statistician. EH and AW-S cowrote the intervention materials. TD codeveloped the study design, cowrote intervention materials and cowrote the manuscript.

**Funding** This work is supported by a grant to TD from the UK Medical Research Council, grant number MC_US_A060_0019.

**Competing interests** None declared.

**Patient consent** Obtained.

**Ethics approval** NHS National Research Ethics Committee (East of England, reference 11/H0305/1).

**Provenance and peer review** Not commissioned; externally peer reviewed.

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
