## [Reviewer comments · BMJ Open]

ARTICLE DETAILS

TITLE (PROVISIONAL)	Study protocol for a randomised, controlled platform trial estimating the effect of autobiographical Memory Flexibility Training (MemFlex) on relapse of recurrent Major Depressive Disorder
AUTHORS	Hitchcock, Caitlin; Gormley, Siobhan; O Leary, Cliodhna; Rodrigues, Evangeline; Chadwick, Isobel; Griffiths, Kirsty; Gillard, Julia; Watson, Peter; Hammond, Emily; Werner-Seidler, Aliza; Dalgleish, Tim

VERSION 1 – REVIEW

REVIEWER	Jonathan P. Stange University of Illinois at Chicago, USA
REVIEW RETURNED	13-Sep-2017

GENERAL COMMENTS	The authors present a well-written manuscript that addresses an important question with a well-designed RCT of autobiographical memory flexibility training for prevention of depressive relapse. The paper might be strengthened by considering some of the comments that follow: Introduction 1. The introduction could benefit from some improved clarity about what constitutes flexibility of memory types, as opposed to the specificity/generalizability of memories. They state that memory flexibility involves “movement between” specific and general memory types. It would be useful to highlight more specifically what this means, and why it would be adaptive to have a variety of memory types, as much of the OGM literature to date would seem to imply that specific memories are always more adaptive (but perhaps this is not necessarily the case). Relatedly, on p. 5, the authors state that memory flexibility is a key predictor of depression, but the papers they cite appear to be about overgeneral memory, not about memory inflexibility. To what extent is there evidence that memory flexibility is also important? Is there evidence that it is more important (in predicting depression relapse) than specific memories?2. The authors state that there is evidence that overgeneral memory is a durable marker for depression that is not mutable by current treatments, but they do not cite a study that shows this. Is there evidence that OGM is stable with treatment? Or is there just evidence that it is naturalistically stable inside and outside of mood episodes?3. Does the MemFlex training always involve utilizing a workbook for the main portion of the intervention, or is this an adaptation for the purposes of the present study?
--

4. It would be useful to note what rate of success the pilot trials of MemFlex had at improving the outcomes (memory specificity, memory flexibility, symptoms or relapse of depression, etc.). It also would be important to note the effect sizes generated from the initial pilot study as this might have implications for the degree of change anticipated in the current study, at least in the MemFlex condition.

Method

5. How will significant cognitive impairment be assessed via self-report?

6. Some participants will be recruited from the existing database of participants. Have any of these participants participated in prior trials of MemFlex (or any other memory specificity training)? Perhaps participants who have done so should be excluded from the current trial to avoid any unintended carry-over effects.

7. The description of how individuals “move between” specific and general memories could be clarified. Perhaps the clarification suggested in the introduction would make it more apparent what this means and why it would be adaptive to do so. In any event, more information about how it is measured would be useful.

8. What procedures are planned if individuals fail to complete the expected twice-weekly sessions over the 4-week period? What proportion of completion will be deemed acceptable for inclusion in the study – will all data be used and any missing data imputed with LOCF?

9. It was not clear to me what the purpose is of “explicitly link[ing] specific event memories with a general theme, and in reverse...”. Is this the portion of MemFlex that is targeting “flexibility” (the ability to switch from one type to the other)? It makes sense that training to become more specific would be useful when initially recalling a general memory. Why would the reverse be adaptive, rather than always being able to recall specific memories being adaptive? This gets back to the broader question of how MemFlex may compare to memory specificity training. Presumably, there is something theoretically more important (or at least as important) about flexibility than about specificity.

10. Do the psychoeducation workbooks take the same amount of time to complete as the MemFlex workbooks? Ideally, the conditions would be matched on as many variables as possible, including time burden.

11. Do the authors plan to assess information about patients’ expectations about which treatment condition is likely to be more effective, or even about expectations about the efficacy of whichever condition they are assigned to? This could be useful for analyses to determine whether treatment expectations (or placebo-like effects) could explain any difference between the conditions that exists in terms of outcome variables such as relapse. For example, if MemFlex seems much more exciting than psychoeducation, it is possible that participant expectations, rather than the memory flexibility mechanisms hypothesized, would explain differences between the conditions.

12. How will participant compliance be measured? How will the researchers know whether participants are actually completing the exercises on schedule as intended (e.g., could electronic books be used that the researchers can see in real time)? Is it possible that participants could just quickly complete the workbook before the mid-treatment and post-treatment assessments (e.g., because of forgetting or lacking motivation)? This could obviously interfere with treatment integrity.

13. The authors state that score on all 3 blocks of the AMT-AI is the main outcome, but would the alternating block really be the primary outcome as this is the true measure of “flexibility” (the ability to switch between retrieval styles in a contextually-appropriate manner)? Regarding measurement of what the “flexible” and “inflexible” individual would look like: presumably a flexible individual would score 100% on all 3 blocks (specific, general, alternating), whereas an “inflexible” individual who always recalls general memories regardless of context would score 0% on either the specific or general block, and perhaps a 50% score on the alternating block (general would be correct 50% of the time on the alternating block). What would a 0% score on the alternating block represent – perhaps a complete mismatch of the alternating pattern with what would be appropriate? It seems that there could be different patterns of responding that could indicate different sub-types of (in)flexibility that might be of interest, in addition to using the total score on the alternating block as the primary indicator of flexibility.

Analyses

14. The authors state that t-tests will be used while covarying for any baseline differences. Presumably this would be an ANCOVA test (not a t-test) if there are any covariates.

15. No hypotheses seem to be stated in the analysis section. Presumably, the authors suspect that MemFlex will be superior to psychoeducation in each of the key outcome variables. Depending on the journal’s requirement/preferences, more specificity (no pun intended) could be included so as to map onto the analyses. For example, in the mixed ANOVAs the authors might hypothesize a significant interaction between group and time such that the MemFlex group shows greater changes over time than psychoeducation group.

16. How will moderators of treatment efficacy be examined in terms of analyses? E.g., interactions between possible moderators and group in survival analyses and regressions, and moderator x group x time interactions in the ANOVAs?

17. What type of analysis will be used for mediation – e.g., tests of indirect effects with bootstrapping?

REVIEWER	Julie Krans Radboud University Nijmegen, the Netherlands Pro Persona Reseach, the Netherlands KU Leuven, Belgium
REVIEW RETURNED	14-Sep-2017

GENERAL COMMENTS	Please inform the reader whether the workbooks are available for other researchers interested in replication research or verifying the content of the applied interventions, and if so, how they can obtain these. Study limitations are not explicitly discussed. If appropriate for a research protocol, it would be helpful to list the limitations that you are aware of along with a brief justification and explanation of the decisions that were made regarding these limitations.
--

REVIEWER	Denise R. Beike, Professor University of Arkansas USA
REVIEW RETURNED	25-Sep-2017

GENERAL COMMENTS	I think this is an important step in the research program and that the MemFlex training could be a very useful adjunct to treatment. It has already been shown to have some efficacy, and this study is the next step in the process. I'll explain my three "no" responses. They are all minor issues. Regarding methodology, I would like to know what instructions will be given to participants after they complete the eight MemFlex sessions. Are they told to try to continue practicing accessing positive and specific memories? How often? Or is no mention made of continuing the exercises on one's own? This could be an important component of the intervention and its long-term efficacy. Regarding statistical analyses, it was unclear to me specifically which baseline differences the researchers plan to examine to use as possible covariates in the analyses. Depression scores, demographics, or what other variables might be considered? I think these need to be spelled out in the document. My final "no" response about supplementary reporting is there because I did not see the dates of the study included in the manuscript. It wasn't clear to me whether the study has already started; looking at the preregistration site at clinicaltrials.gov, it looks like it began in 2016. Can this be clarified? Other than these minor issues, the study seems well-justified, the methodology sound, the protection against biases rigorous and well thought-out, and the analyses appropriate.
---

VERSION 1 – AUTHOR RESPONSE

Reviewer: 1

Reviewer Name: Jonathan P. Stange

Institution and Country: University of Illinois at Chicago, USA Please state any competing interests or state 'None declared': None declared

The authors present a well-written manuscript that addresses an important question with a well-designed RCT of autobiographical memory flexibility training for prevention of depressive relapse. The paper might be strengthened by considering some of the comments that follow:

Introduction

1. The introduction could benefit from some improved clarity about what constitutes flexibility of memory types, as opposed to the specificity/generalizability of memories. They state that memory flexibility involves “movement between” specific and general memory types. It would be useful to highlight more specifically what this means, and why it would be adaptive to have a variety of memory types, as much of the OGM literature to date would seem to imply that specific memories are always more adaptive (but perhaps this is not necessarily the case). Relatedly, on p. 5, the authors state that memory flexibility is a key predictor of depression, but the papers they cite appear to be about overgeneral memory, not about memory inflexibility. To what extent is there evidence that memory flexibility is also important? Is there evidence that it is more important (in predicting depression relapse) than specific memories?

Response: Thank you for making this apparent. We have now edited the Introduction to improve clarity. In particular, we now state that ‘Memory flexibility refers to the ability to successfully navigate the autobiographical information database to deliberately retrieve memories varying in detail and emotional valence.’ (page 4). The discussion of why it is adaptive to recall the different memory types has also been expanded (pages 4-5).

Although there is evidence that in addition to reduced specificity, depressed individuals experience difficulty with the intentional recall of general memories, and also an impaired ability to alternate between specific and general memories, the predictive effects of impaired generality or flexibility more broadly have yet to be examined. This has now been stated in the Introduction (page 5).

2. The authors state that there is evidence that overgeneral memory is a durable marker for depression that is not mutable by current treatments, but they do not cite a study that shows this. Is there evidence that OGM is stable with treatment? Or is there just evidence that it is naturalistically stable inside and outside of mood episodes?

Response: This statement has been clarified to reflect that overgeneral memory remains stable between depressive episodes (Williams et al., 2007; Sumner et al., 2013), and that the impact of therapy on OGM has not been well explored (page 6), as to our knowledge, only one study has explicitly examined this relationship (Williams et al., 2000).

3. Does the MemFlex training always involve utilizing a workbook for the main portion of the intervention, or is this an adaptation for the purposes of the present study?

Response: The intervention is always delivered via a workbook, and the Method has been clarified to reflect that the intervention is delivered as per the initial pilot study, and our recently completed treatment trial (page 9).

4. It would be useful to note what rate of success the pilot trials of MemFlex had at improving the outcomes (memory specificity, memory flexibility, symptoms or relapse of depression, etc.). It also would be important to note the effect sizes generated from the initial pilot study as this might have implications for the degree of change anticipated in the current study, at least in the MemFlex condition.

Response: We have now inserted the effect sizes from the uncontrolled trial for effects of MemFlex on overgeneral memory, rumination, problem solving, and cognitive avoidance (page 7).

Method

5. How will significant cognitive impairment be assessed via self-report?

Response: At eligibility screening, participants will be asked if they have ever experienced an intellectual disability or traumatic brain injury. This has now been added to the Method (page 8).

6. Some participants will be recruited from the existing database of participants. Have any of these participants participated in prior trials of MemFlex (or any other memory specificity training)? Perhaps participants who have done so should be excluded from the current trial to avoid any unintended carry-over effects.

Response: We agree that this is important, and participation in any prior autobiographical memory-based intervention is an exclusion criterion for this study. This has now been added to Method (page 8).

7. The description of how individuals “move between” specific and general memories could be clarified. Perhaps the clarification suggested in the introduction would make it more apparent what this means and why it would be adaptive to do so. In any event, more information about how it is measured would be useful.

Response: The Method has now been edited to state that individuals must move between memory types by recalling both specific and general memories to the same cue (page 11). In addition to the edits made to the Introduction (described in response to point 1), we have also expanded on the importance of both specific and general memories in daily cognition (page 5). We hope that these edits will adequately clarify this issue, but we of course stand ready to make any further revisions that you feel are necessary to clarify this point.

8. What procedures are planned if individuals fail to complete the expected twice-weekly sessions over the 4-week period? What proportion of completion will be deemed acceptable for inclusion in the study – will all data be used and any missing data imputed with LOCF?

Response: The progress telephone call will determine how many sessions have been completed, and a revised date for the post-intervention assessment will be set in collaboration with the participant. In our prior experience with the MemFlex programme, total completion time has not exceeded eight weeks (average of 4-6 weeks). The facilitator will call back the week before the scheduled post-intervention assessment to confirm that the workbook will be completed in time for the appointment. This information has now been clarified in the Method (page 12).

There is no minimum number of completed sessions for inclusion in the study, in line with intent-to-treat analysis. Any missing data will be accounted for using multiple imputation, outlined in the Statistical Analysis Plan. Further detail on the imputation method to be used has now been added to the Statistical Analysis Plan (page 15). Although no per-protocol analyses are planned, we have added a line stating that any exploratory per-protocol analyses will use data from individuals who have completed six out of the eight workbook sessions (page 15).

9. It was not clear to me what the purpose is of “explicitly link[ing] specific event memories with a general theme, and in reverse...”. Is this the portion of MemFlex that is targeting “flexibility” (the ability to switch from one type to the other)? It makes sense that training to become more specific would be useful when initially recalling a general memory. Why would the reverse be adaptive, rather than always being able to recall specific memories being adaptive? This gets back to the broader question of how MemFlex may compare to memory specificity training. Presumably, there is something theoretically more important (or at least as important) about flexibility than about specificity.

Response: As we have now expanded on in the Introduction, depression is associated with the impaired retrieval of both specific and general memories, and both memory types are important in daily cognition (page 5). Improving recall of general memories may thereby impact daily functioning. In addition, explicitly training improved recall of positive, general memories (e.g., of kindness, determination, caring relationships) may help to support re-initiation of the overly general, positive self-judgements that are associated with mental health (for review see Taylor & Brown, 1988), and thereby counter the negative overgeneralisations that characterise depression. This assumption is consistent with wider literature regarding the role of general memories in shaping self-judgements (Klein et al., 2001, 2002). This rationale is now elaborated on in the Introduction (page 6).

10. Do the psychoeducation workbooks take the same amount of time to complete as the MemFlex workbooks? Ideally, the conditions would be matched on as many variables as possible, including time burden.

Response: Time burden is matched between conditions. This has now been explicitly stated on page 11.

11. Do the authors plan to assess information about patients' expectations about which treatment condition is likely to be more effective, or even about expectations about the efficacy of whichever condition they are assigned to? This could be useful for analyses to determine whether treatment expectations (or placebo-like effects) could explain any difference between the conditions that exists in terms of outcome variables such as relapse. For example, if MemFlex seems much more exciting than psychoeducation, it is possible that participant expectations, rather than the memory flexibility mechanisms hypothesized, would explain differences between the conditions.

Response: Thank you for this comment. An oversight on our part saw that we failed to report that we were administering the Credibility Expectancy Questionnaire – Patient Version (Deville & Borkovec, 2000). This has now been added to the method (page 12).

12. How will participant compliance be measured? How will the researchers know whether participants are actually completing the exercises on schedule as intended (e.g., could electronic books be used that the researchers can see in real time)? Is it possible that participants could just quickly complete the workbook before the mid-treatment and post-treatment assessments (e.g., because of forgetting or lacking motivation)? This could obviously interfere with treatment integrity.

Response: You raise an interesting point, and internet-based workbooks is something that we have considered, however we opted to keep the workbooks in hardcopy due to frequently-reported technical issues and difficulty with engagement in internet-delivered interventions, along with participant feedback from our pilot study indicating a preference for a hardcopy book. During the initial session, participants are explicitly instructed that there needs to be time between the sessions to consolidate the newly learnt skill, and the progress phone call allows the opportunity of delaying the post-assessment to allow the individual time to complete the workbook. As currently collected data indicates that participants do make use of this option, cramming before assessments seems unlikely. We have noted in the Strengths and Limitations section (page 3) that use of hardcopy workbooks limits our ability to track workbook completion in real-time.

13. The authors state that score on all 3 blocks of the AMT-AI is the main outcome, but would the alternating block really be the primary outcome as this is the true measure of “flexibility” (the ability to switch between retrieval styles in a contextually-appropriate manner)? Regarding measurement of what the “flexible” and “inflexible” individual would look like: presumably a flexible individual would score 100% on all 3 blocks (specific, general, alternating), whereas an “inflexible” individual who always recalls general memories regardless of context would score 0% on either the specific or general block, and perhaps a 50% score on the alternating block (general would be correct 50% of the time on the alternating block). What would a 0% score on the alternating block represent – perhaps a complete mismatch of the alternating pattern with what would be appropriate? It seems that there could be different patterns of responding that could indicate different sub-types of (in)flexibility that might be of interest, in addition to using the total score on the alternating block as the primary indicator of flexibility.

Response: As clarified above, depression is associated with impaired recall of specific memories, impaired recall of general memories (Dalglish et al., 2007), and impaired ability to alternate between the memory types (Dritschel et al., 2014). As such, the total score will allow an index of the overall ability to successfully navigate the autobiographical information store and deliberately retrieve all memory types. Exploratory analyses can be used to explore the flexibility subtypes you suggest, and this has been added to the statistical analyses section (page 15).

Analyses

14. The authors state that t-tests will be used while covarying for any baseline differences. Presumably this would be an ANCOVA test (not a t-test) if there are any covariates.

Response: This has now been changed (page 14).

15. No hypotheses seem to be stated in the analysis section. Presumably, the authors suspect that MemFlex will be superior to psychoeducation in each of the key outcome variables. Depending on the journal’s requirement/preferences, more specificity (no pun intended) could be included so as to map onto the analyses. For example, in the mixed ANOVAs the authors might hypothesize a significant interaction between group and time such that the MemFlex group shows greater changes over time than psychoeducation group.

Response: Hypotheses are stated at the end of the Introduction (page 8).

16. How will moderators of treatment efficacy be examined in terms of analyses? E.g., interactions between possible moderators and group in survival analyses and regressions, and moderator x group x time interactions in the ANOVAs?

Product variables will be computed to explore moderation effects using linear regression. This detail has now been added to page 14.

17. What type of analysis will be used for mediation – e.g., tests of indirect effects with bootstrapping? That is correct- PROCESS will be used to test indirect effects using bootstrapped samples. This has now been added to page 15.

Reviewer: 2

Reviewer Name: Julie Krans

Institution and Country: Radboud University Nijmegen, the Netherlands; Pro Persona Reseach, the Netherlands; KU Leuven, Belgium Please state any competing interests or state 'None declared':

None declared

Comment: Please inform the reader whether the workbooks are available for other researchers interested in replication research or verifying the content of the applied interventions, and if so, how they can obtain these.

Response: Once the trial has been completed, and any potential efficacy has been established, the workbooks will be made available online. Until then, they are available on request from the authors. This information has now been added to the manuscript (page 11).

Comment: Study limitations are not explicitly discussed. If appropriate for a research protocol, it would be helpful to list the limitations that you are aware of along with a brief justification and explanation of the decisions that were made regarding these limitations.

Response: Thank you for highlighting this oversight. We have now added expanded the discussion of limitations in the Strengths and Limitations section (page 3). We have also made edits throughout the manuscript to more comprehensively reflect the reasoning behind methodological decisions.

Reviewer: 3

Reviewer Name: Denise R. Beike, Professor Institution and Country: University of Arkansas, USA Please state any competing interests or state 'None declared': None declared

Comment: I think this is an important step in the research program and that the MemFlex training could be a very useful adjunct to treatment. It has already been shown to have some efficacy, and this study is the next step in the process.

I'll explain my three "no" responses. They are all minor issues.

Regarding methodology, I would like to know what instructions will be given to participants after they complete the eight MemFlex sessions. Are they told to try to continue practicing accessing positive and specific memories? How often? Or is no mention made of continuing the exercises on one's own?

This could be an important component of the intervention and its long-term efficacy.

Response: We agree that this is an important issue. There are additional exercises at the end of the workbook that the participant can choose to do after the study has ended. While participants are encouraged to continue to apply their new skills in their everyday life, completion of the additional workbook exercises is optional, and we are unsure how many participants will elect to continue practicing. This information has now been added to the Method (page 10).

We agree that continued practice would promote long-term efficacy, and have noted this as a limitation in the Strengths and Limitations section (page 3). We will also keep this in mind in further revisions of the intervention (e.g., potential addition of booster sessions).

Comment: Regarding statistical analyses, it was unclear to me specifically which baseline differences the researchers plan to examine to use as possible covariates in the analyses. Depression scores, demographics, or what other variables might be considered? I think these need to be spelled out in the document.

Response: Any baseline differences in the cognitive target, symptom severity on BDI-II, concurrent treatment, or demographics will be covaried in analyses. This information has now been added to the Statistical Analysis section (page 14).

Comment: My final "no" response about supplementary reporting is there because I did not see the dates of the study included in the manuscript. It wasn't clear to me whether the study has already started; looking at the preregistration site at clinicaltrials.gov, it looks like it began in 2016. Can this be clarified?

Response: Thank you for highlighting this inconsistency. The trial protocol was registered online on 25 November 2015, with recruitment planned to open in January 2016. Recruitment began on 18 January 2016. We have now clarified this information in the Trial Registration Section (page 16).

Comment: Other than these minor issues, the study seems well-justified, the methodology sound, the protection against biases rigorous and well thought-out, and the analyses appropriate.

Response: Thank you for your helpful comments.

We appreciate the time and effort each of you has given to this manuscript. We hope that these edits clarify the issues you have raised, but we of course stand ready to make any further revisions that you feel are necessary.

VERSION 2 – REVIEW

REVIEWER	Jonathan P. Stange University of Illinois at Chicago, USA
REVIEW RETURNED	31-Oct-2017

GENERAL COMMENTS	The authors have done a commendable job at responding to the initial comments. I have no further suggestions to make.
---

REVIEWER	Julie Krans Radboud University Nijmegen, the Netherlands Pro Persona Research, the Netherlands KU Leuven, Belgium
REVIEW RETURNED	15-Nov-2017

GENERAL COMMENTS	The authors have addressed all my queries and I have no further comments. I wish them good luck with the study and am looking forward to hearing about the results.
---

REVIEWER	Denise Beike, Professor University of Arkansas, USA
REVIEW RETURNED	11-Nov-2017

GENERAL COMMENTS	I had a favorable opinion of the previous version of this manuscript. After having read the other reviewers' suggestions in addition to my own, I am satisfied that the authors have addressed all major concerns.
--